# Mathematical Modeling and Design of Parboiled Paddy-Impinging Stream Dryer Using the CFD-DEM Model

**DOI:** 10.3390/foods13101559

**Published:** 2024-05-16

**Authors:** Thanit Swasdisevi, Wut Thianngoen, Somkiat Prachayawarakorn

**Affiliations:** 1Division of Thermal Technology, School of Energy, Environment and Materials, King Mongkut’s University of Technology Thonburi, 126 Pracha u-tid Road, Tungkru, Bangkok 10140, Thailand; thanit.swa@kmutt.ac.th; 2Division of Energy Technology, School of Energy, Environment and Materials, King Mongkut’s University of Technology Thonburi, 126 Pracha u-tid Road, Tungkru, Bangkok 10140, Thailand; wut.thia@kmutt.ac.th; 3Department of Chemical Engineering, Faculty of Engineering, King Mongkut’s University of Technology Thonburi, 126 Pracha u-tid Road, Tungkru, Bangkok 10140, Thailand

**Keywords:** CFD-DEM, ISD, parboiled paddy, residence time distribution

## Abstract

Impinging stream dryers (ISDs) are effective for removing moisture from particulate materials because of the complex multiphase transport of air particles in ISDs. Nowadays, computational techniques are powerful to simulate multiphase flows, including dilute and dense-phase gas–solid flows and hence, the use of a reliable computational model to simulate the phenomena and design a dryer has recently received more attention. In this study, computational fluid dynamics, combined with the discrete element method (CFD-DEM) and falling drying rate model, were used to predict the multiphase transport phenomena of parboiled paddy in a coaxial ISD. The design of an impinging stream pattern for improving residence time in a drying chamber of ISD was also investigated. The results showed that the CFD-DEM, in combination with the falling drying model, could be well-utilized to predict the particle motion behavior and lead to more physically realistic results. The predicted change of moisture content in parboiled paddy was in good agreement with the experimental data for 17 cycles of drying. Although the prediction of mean residence time was lower than the experimental data, the predicted mean residence time was a similar trend to the experimental data. For ISD design, the simulation revealed that the use of two stages of impinging stream region (two streams collide at the top of the drying chamber at the first stage and then the gas particles flow on the incline floor to collide with the other stream at second stage) in a drying chamber could increase the residence time approximately 75% and reduce drying cycle from 17 to 10 cycles when it was considered at the same final moisture content.

## 1. Introduction

The impinging stream technique reveals the potential for drying high-moisture particulate materials. Two streams of hot air or more are entered into a drying chamber at high velocities on opposite sides, while one stream of hot air carries a material to be dried. The opposite streams collide at the impingement zone, leading to a high shear rate and intense turbulence; hence, mass, momentum, and heat transfer rates between hot air and particles increase. In addition, the particles penetrate the opposite stream, which can help prolong the particle’s residence time. These phenomena are important in ISD because they can enhance the interphase heat and mass transfer [1]. Kudra and Mujumdar [2,3] investigated various impinging stream techniques for drying particulate materials. They found that the performance of ISD for removing surface moisture of particles was better than that of other dryers, such as fluidized beds and pneumatic and spray dryers. Nimmol et al. [4] investigated a coaxial ISD for paddy drying in terms of drying air temperature, paddy feed rate, impinging distance and energy consumption. The results revealed that the drying rate of ISD was higher than that of fluidized beds and spouted beds approximately 40 and 250 times, respectively. Such information indicates an important advantage of ISD over the other dryer types. However, the residence time of particles in ISD is very short; hence, the drying of high-moisture particulate seeds would be used for many drying cycles. Kumklam et al. [5] investigated parboiled paddy drying using ISD. They found that the ISD used 16–22 drying cycles to dry the parboiled paddy to reduce the moisture content from 50% d.b. to 18–19% d.b., depending upon the operating condition. Since particles spend a very short time in ISD, it is very difficult to experimentally study the multiphase transport phenomena. In the present day, the computational techniques are powerful enough to simulate multiphase flow; hence, the use of a powerful computer to simulate the complex multiphase flow phenomena in the impinging stream chamber instead of a tedious experimental route is desirable in order to assist in designing an ISD. Hosseinalipour and Mujumdar [6,7] studied particle drying characteristics and heat transfer in ISD using Eulerian–Lagrangian model. The results showed that the relative velocity between particles and the drying medium and the turbulence flow in the impingement zone could enhance the drying process. However, the effect of particle–particle interactions was not investigated in this study. Choicharoen et al. [8] presented a three-dimensional computational fluid dynamics model to study the multiphase transport phenomena in ISD using the Eulerian–Lagrangian model with standard k-ε and realizable k-ε turbulence model. It was revealed that the particle moisture content, air humidity ratio, and particle mean residence time of simulation were approximately different from the experimental data. In addition, the realizable k-ε turbulence model performed better than the k-ε turbulence model to simulate the particle mean residence time and moisture reduction. However, the model did not consider the particle–particle interactions. The CFD models used to simulate the multiphase flow within an ISD did not include the particle–particle interactions [8,9], leading to unrealistic particle motion; hence the simulated results were quite different from the experimental data. 

Du et al. [10,11] presented a simulation of the particle motion behavior in the drying process in an ISD using the modified direct simulation Monte Carlo (DSMC) method. They found that the oscillatory motion of particles could increase the mean residence time of particles and intensify the drying process. The performance of the dryer increased with an increase in the impinging distance. However, the hard-sphere model with a simple binary collision was used to simulate particle motion in this work. In addition, the hard-sphere model was suitable for a lower particle concentration in the flow field [12,13]. The discrete element method (DEM) was an alternative method for simulating a flow field with a dense phase of particles, including particle–particle interaction in terms of collisions. Many researchers [14,15,16,17,18,19] have applied CFD-DEM to study the gas–solid flow field in a spouted bed, fluidized bed, and impinging stream. They found that the simulated results in terms of gas and particle motion agreed well with the experimental results. Khomwachirakul et al. [18] used a CFD-DEM and constant drying rate model to investigate multiphase transport phenomena within a coaxial ISD. The results showed that CFD-DEM provided more physically realistic results of particle motion compared to CFD. The simulated particle moisture content mean and residence time were quite close to the experimental data. However, the drying was carried out in only one cycle, and the constant drying rate model was applied to the CFD-DEM model, whereas the mechanism of parboiled paddy drying was a falling drying rate, and ISD used many drying cycles to dry the parboiled paddy for each condition.

As mentioned above, most mathematical models developed for ISD have been applied to dry material at a constant rate. The model has rather been limited in the literature for the falling rate period. In the present work, CFD-DEM, including the falling rate model, was therefore developed and applied to investigate the parboiled paddy movement as well as the drying kinetics of parboiled paddy in an ISD. The developed model was validated with the experimental data in terms of mean grain moisture content, mean outlet grain temperature and mean residence time. In addition, the developed model was applied to design an ISD to reduce the number of drying cycles and increase the mean residence time. 

## 2. Materials and Methods

The experimental results used for validating the mathematical model were obtained from Kumklam et al. [5]. In their work, the parboiled paddy was dried by a pilot scale ISD with a diameter of 1.05 m and height of 1.1 m. The inlet pipe of the drying chamber was 0.038 m in diameter. The impinging distance could be varied between 5 and 17.5 cm. The drying air velocity varied between 15 and 25 m/s. The parboiled paddy feed rates of 160 and 320 kg/s were used. The details of experimental conditions are given in Table 1. Before starting the experiment, the drying system was warmed up until the temperature at the drying chamber was close to the desired drying temperature. The parboiled paddy was fed into the drying chamber and collected at the bottom of ISD, as shown in Figure 1. The sample was then dried again by ISD, which was called the drying cycle in this work. In this work, the sample was dried for at least 17 drying cycles. 

In the present work, the falling drying rate model was included in CFD-DEM to investigate the particle motion behavior as well as the drying kinetics of the parboiled paddy in an ISD. The turbulence model of the gas phase used in the simulation was a realizable *k-*ε model [8], and the soft sphere model was applied to particle–particle and particle–wall collisions [18].

### 2.1. Mathematical Equations

#### 2.1.1. Gas Phase Motion

Gas motion was simulated three-dimensionally with a continuity equation, including the momentum equation and the turbulence model was used to calculate gas motion. The equations of gas motion are written as follows:

Continuity equation:(1)$$\frac{\partial }{\partial t}\left(\alpha_{c}\rho_{c}\right)=\frac{\partial }{\partial x_{i}}\left(\alpha_{c}\rho_{c}u_{i}\right)+S_{mass}$$
(2)$$S_{mass}=\frac{1}{V_{finite.volume}}\sum_{a=1}^{n}{\dot{m}}_{p,a}$$
(3)$$\alpha_{c}=\frac{V_{c}}{V_{finite.volume}}$$
where $S_{mass}$ is the volumetric mass source, which reflects the coupling between the particle and gas phases [12]: 

Momentum equation: (4)$$\frac{\partial }{\partial t}\left(\alpha_{c}\rho_{c}u_{j}\right)=\frac{\partial }{\partial x_{j}}\left(\alpha_{c}\rho_{c}u_{i}u_{j}\right)+\alpha_{c}\frac{\partial P}{\partial x_{j}}+\frac{\partial }{\partial x_{j}}(\alpha_{c}\tau_{ij})+\alpha_{c}\rho_{c}g+S_{momentum}$$
(5)$$\tau_{ij}=(\mu +\mu_{t})(\frac{\partial u_{i}}{\partial x_{j}}+\frac{\partial u_{j}}{\partial x_{i}})$$
(6)$$S_{momentum}=-\frac{1}{V_{finite.volume}}\sum_{a=1}^{n}\frac{\beta \cdot V_{p,a}}{1-\alpha_{c}}(u_{c}-u_{p})$$
(7)$$u_{c}=\sqrt{{u^{2}}_{i}}$$
(8)$$u_{p}=\sqrt{{u^{2}}_{p,i}}$$
where $S_{momentum}$ is the volumetric gas–particle interaction force, which reflects the coupling between the particle and gas phases [19]. The interphase momentum exchange coefficient (β) based on the soft-sphere model [12] was calculated from the Wen and Yu correlation [12,20] as follows:(9)$$\beta =\frac{3}{4}C_{D}\frac{\alpha_{c}(1-\alpha_{c})}{d_{p}}\rho_{c}\left|u_{c}-u_{p}\right|{\alpha_{c}}^{-2.7}\;\;\;\;\mathrm{for}\;\alpha_{c}>0.8$$
$$\beta =\frac{(1-\alpha_{c})\mu }{\alpha_{c}d_{p}^{2}}[150(1-\alpha_{c})+1.75\alpha_{c}\rho_{c}d_{p}\frac{\left|u_{c}-u_{p}\right|}{\mu }]\;\mathrm{for}\;\alpha_{c}\leq 0.8$$

The drag coefficient ($C_{D}$) of parboiled paddy was calculated by non-spherical particles [21] as follows: (10)CD=24Resph(1+b1Resphb2)+b3Resphb4+Resph
$$\begin{array}{l} b_{1}=\exp(2.3288-6.4581\varphi +2.4486\varphi^{2})\\ b_{2}=0.0964+0.5565\varphi \\ b_{3}=\exp(4.905-13.8944\varphi +18.4222\varphi^{2}-10.2599\varphi^{3})\\ b_{4}=\exp(1.4681+12.2584\varphi -20.7322\varphi^{2}+15.8855\varphi^{3}) \end{array}$$

The shape factor ($\varphi =\frac{s}{S}$) is the ratio of spherical surface area and real surface area at the same particle volume. The Reynolds number was written as follows:(11)Re=u→c−u→pρcαcdpμ

Turbulence model:

The realizable *k-*ε model was used to calculate the gas-phase turbulence, and the equations are as follows [12,22,23,24]:(12)$$\frac{\partial }{\partial t}\left(\alpha_{c}\rho_{c}k\right)=\frac{\partial }{\partial x_{i}}\left(\alpha_{c}\rho_{c}ku_{i}\right)+\frac{\partial }{\partial x_{j}}\left(\alpha_{c}(\mu +\mu_{t})\frac{\partial k}{\partial x_{j}}\right)+\alpha_{c}G_{k}-\alpha_{c}\rho_{c}\varepsilon$$
(13)$$\frac{\partial }{\partial t}\left(\alpha_{c}\rho_{c}\varepsilon \right)=\frac{\partial }{\partial x_{i}}\left(\alpha_{c}\rho_{c}\varepsilon u_{i}\right)+\frac{\partial }{\partial x_{j}}\left(\alpha_{c}(\mu +\frac{\mu_{t}}{1.2})\frac{\partial \varepsilon }{\partial x_{j}}\right)-1.9\alpha_{c}\rho_{c}\frac{\varepsilon^{2}}{k+\sqrt{v\varepsilon }}$$

The realizable *k-***ε** model is useful for high turbulence systems.

#### 2.1.2. Particle Motion

Particle motion was simulated by simultaneously applying Newton’s second law of motion to each individual particle and taking into account the gravitational force ($\vec{F_{g}}$), gas–solid drag force (${\vec{F}}_{D}$) and contact forces between particles (${\vec{F}}_{C}$) modeled with the soft-sphere model.

The drag force ${\vec{F}}_{D}$ is defined as [12,18]:(14)$$\vec{F_{D}}=(\frac{\beta }{1-\alpha_{c}}({\vec{u}}_{c}-{\vec{u}}_{p}))V_{p}$$

Contact force was composed of a spring, dash-pot and friction slider [12,14]. The contact force (${\vec{F}}_{C}$) was divided into a normal component (${\vec{F}}_{Cn}$) and a tangential component (${\vec{F}}_{Ct}$) as follows: (15)$${\vec{F}}_{C}={\vec{F}}_{Cn}+{\vec{F}}_{Ct}$$

The normal contact force and the tangential contact force can be calculated according to Khomwachirakul et al. [18]. Particle motion was simulated by simultaneously applying Newton’s second law of motion to each individual particle, and it can be expressed as [16,17]:(16)$$m_{p}\frac{d\vec{u_{p}}}{dt}=\vec{F_{g}}+\vec{F_{D}}+\vec{F_{C}}$$

The empirical parameters that must be specified in DEM include spring constant, friction coefficient and restitution coefficient. Tsuji et al. [14] and Kaneko et al. [25] revealed that the spring constant does not significantly affect the particles’ motion; hence, the use of a smaller spring constant could shorten the computation time. In previous work, the appropriate spring constant was in the range of 800–1000 N/m [17,26,27,28]; hence, the spring constant was assumed to be 1000 N/m. Wongbubpa et al. [29] found that the restitution coefficients of parboiled paddy with parboiled paddy and stainless steel were 0.6 and 0.6, respectively. The friction coefficient of parboiled paddy with parboiled paddy was 0.3 [29], and the friction coefficient of parboiled paddy with stainless steel was 0.35 [30]. Those values were chosen for DEM simulation.

#### 2.1.3. Energy Equation

In an ISD, the convective heat transfer between gas and particles was the main mechanism of heat transfer, and a model of convective heat transfer can be calculated by the energy balance between particles and gas phases. The energy equation for the gas phase can be expressed as [18,31,32]:(17)$$\frac{\partial }{\partial t}(\alpha_{c}\rho_{c}c_{p}T_{c})=\frac{\partial }{\partial x_{i}}(\alpha_{c}\rho_{c}c_{p}u_{i}T_{c})+\frac{\partial }{\partial x_{i}}(\alpha_{c}k_{c}\frac{\partial T_{c}}{\partial x_{i}})+S_{energy}$$
(18)$$S_{energy}=-\sum_{a=1}^{n}\frac{6(1-\alpha_{c})}{d_{p}}h(T_{c}-T_{p,a})$$
where *S_energy_* is the volumetric heat source in the computational domain of the gas phase, which describes the energy transferring from the drying air to the particles [18,25]:

The convective gas-to-particle heat transfer $\left(h\right)$ was calculated as follows: (19)Nu=2+0.6Re1/2Pr1/3
(20)$$\mathrm{Where}\;\mathrm{Nu}=\frac{hd_{p}}{k_{c}}$$
(21)$$\mathrm{and}\;\mathrm{Pr}=\frac{c_{p}\mu }{k_{c}}$$

The particle temperature can be calculated by the following equation:(22)$$m_{p}c_{p}\frac{dT_{p}}{dt}=hA_{p}\left(T_{g}-T_{p}\right)+\frac{dm_{p}}{dt}h_{\mathrm{fg}}$$

#### 2.1.4. Moisture Transfer Equation

In this work, the falling drying rate model was applied to the CFD-DEM model to predict moisture content during the drying process in an ISD. The moisture transfer within parboiled paddy was controlled only by liquid diffusion. The liquid diffusion equation can be written as [33]:(23)$$\frac{\partial M_{p}}{\partial t}=D_{eff}(\frac{\partial^{2}M_{p}}{\partial r^{2}}+\frac{2}{r}\frac{\partial M_{p}}{\partial r}),0\leq r\leq R_{p}$$

Initial condition:(24)$$M_{p}=M_{0},t=0\&0\leq r\leq R_{p}$$

Boundary condition:(25)$$\frac{\partial M_{p}}{\partial r}=0,t>0\&r=0$$
(26)$$D_{eff}\frac{\partial M_{p}}{\partial r}=h_{m}(M_{e}-M_{s}),t>0\&r=R_{p}$$

The effective diffusion coefficient and equilibrium moisture content were calculated as follows:(27)$$D_{eff}=D_{0}\cdot e^{\frac{-E_{a}}{R\cdot T_{p}}}$$
(28)$$M_{e}=0.01{\left|\frac{In(1-RH)}{-3.146\cdot 10^{-6}\cdot T_{c}}\right|}^{1/2.464}$$

The moisture transfer rate between the particle and gas was then calculated by:(29)$$\frac{\partial m_{p}}{\partial t}=\rho_{d}\cdot V_{p}\cdot \frac{\partial M_{p}}{\partial t}$$

It is noted that the moisture transfer equation was included in the CFD-DEM model by using the user-defined function of ANSYS FLUENT software (student version 2022R1) [22].

## 3. Model Assumptions and Simulation Cases

Figure 1 shows the geometry of ISD used to simulate the drying process of parboiled paddy for 17 cycles. The simulated results were compared to the experimental data [5]. The assumptions used in this simulation were as follows: particles (parboiled paddy) had a uniform spherical shape with a diameter of 0.0039 m during the drying process [5]; particle–particle and particle–wall heat transfer was neglected; negligible rotation of particles during their movement. Air was an ideal gas, and the properties of air, including viscosity, thermal conductivity and heat capacity, were specified as a function of temperature [34]; heat loss through the drying chamber wall was neglected. The coupled CFD-DEM equations, in combination with the falling drying rate model, were solved using the finite volume method through ANSYS FLUENT software (student version 2022R1). The particle equations were solved using an implicit discretization scheme, while the SIMPLE algorithm was used to calculate the coupling between the pressure and velocity in the gas phase. In addition, the spatial discretization for the conservation equations was calculated using the power-law scheme. In this simulation, the gas-phase time step was 1 × 10^−3^ s while the particle-phase time step was 5 × 10^−5^ s, determined by $\Delta t\leq \pi /5\sqrt{m_{p}/k}$ as recommended by Tsuji et al. [14].

The properties of the particle and gas are presented in Table 2 and Table 3, respectively. Figure 2 and Figure 3 present the design of ISD geometry for improving residence time. In Figure 2, the inlet pipe of an ISD chamber was increased to investigate the particles’ behavior and residence time. The design of ISD geometry with two impinging stream regions in a drying chamber (two-stage ISD) for improving particle residence time is presented in Figure 3. The simulation conditions used in this work were as follows: drying air temperature was 190 °C; air velocity was 15, 20 and 25 m/s; the impinging distance was 5, 17.5 and 25 cm; particle feed rate was 160 and 320 kg_dry_/h. 

## 4. Boundary and Initial Conditions

Initially, the particle moisture content and temperature were set at 0.5 d.b. and 28 °C, respectively. The gas temperature was set at 25 °C, and pressure was assumed to be at an atmospheric pressure while the turbulence intensity was set at 5%. The particle velocity was assumed to be zero at the inlet, and the gas velocity was set as the simulation condition. For the gas phase, the side wall was treated as a non-slip wall, and the outlet was assumed to be at atmospheric pressure. In addition, the heat loss through the side wall of an ISD was neglected. 

## 5. Results and Discussion

Prior to the start of simulation data collection, the CFD-DEM was run in combination with the falling drying rate model until the number of inlet particles in the chamber equaled the number of outlet particles from the chamber (steady state). The probability of particle distribution in the chamber had an oscillatory behavior. At the start point of the oscillation, the same pattern was defined as stable time (marginally stable [40]). Table 1 shows the stable time of ISD at various conditions.

### 5.1. Simulation of Drying Process in an ISD

The impinging stream was simulated with CFD-DEM in combination with the falling drying rate model, and the simulated results were verified using the experimental data [5,41]. In addition, the CFD-DEM, in combination with the falling drying rate model, was adopted to design an ISD to reduce the drying cycles and increase the particle mean residence time.

#### 5.1.1. Effect of Air Velocity 

Figure 4 shows the simulated particle motion behavior in the ISD at different inlet air velocities (15, 20 and 25 m/s), a feed rate of 320 kg_dry_/h, a temperature of 190 °C and an impinging distance of 5 cm. The particles moved faster along the axial direction with increasing inlet air velocity due to an increase in the drag force and were uniformly distributed within the right-hand-side conveying pipe. When the particles reached the impingement zone, some particles penetrated the opposite stream because of their momentum. Meanwhile, some particles spread out from the impingement zone to all directions in the drying chamber due to the particle–particle collision. It was noted that the concentration of particles within the opposite stream (left-hand side) at the outlet zone increased with the decrease in inlet air velocity due to the lower air velocity, leading to lower particle momentum. Hence, the particles could not penetrate the conveying pipe where the particles were first introduced.

Figure 5 shows the simulated particle motion in the impingement zone at various inlet air velocities, a feed rate of 320 kg_dry_/h, a temperature of 190 °C and an impinging distance of 5 cm. To observe the oscillation of particles in the impinging stream, 90 particles were fed into the drying chamber at the right-hand-side inlet every 0.03 s. At velocities of 15 m/s and 20 m/s, the particles would move to the opposite stream and then spread out from the impingement zone to the drying chamber. This is because the momentum of the particles at both velocities was not high enough, and hence, the particles could not move back into the right-hand-side inlet, resulting in a number of oscillations being zero. However, the particles could move back into the right-hand-side inlet at a velocity of 25 m/s before they spread out from the impingement zone to the drying chamber due to the higher momentum of the particles. However, there was only one oscillation at this velocity. 

Figure 6a shows a comparison between the simulated moisture content of parboiled paddy particles and experimental data for various air velocities at a feed rate of 320 kg_dry_/h, a temperature of 190 °C and an impinging distance of 5 cm. The particle moisture content decreased with an increase in the drying cycle and decreased with an increase in air velocity due to an increase in evaporated water at the particle surface and particle residence time, as shown in Table 4. During the first period of the drying cycle, the simulated moisture contents were higher than the experimental data because the particle exiting from the drying chamber used a time of approximately 0.3–0.4 s to drop into a storage bag in the experiment, and hence, water could be evaporated from the particle. After nine drying cycles, both the simulated results and experimental data were quite close to each other. The discrepancy between the simulated results and experimental data was approximately 4.4–5.6%. 

Figure 6b shows a comparison between the simulated particle temperature and experimental data for various air velocities at the same feed rate and temperature, as previously mentioned in Figure 6a. As expected, the particle temperature increased with an increase in the drying cycle. The particle temperature increases with an increase in air velocity because a higher air velocity leads to higher convective heat transfer to grains, and a higher particle means residence time. The simulated particle temperature was different from the experimental data, approximately 1.8–3.9%.

#### 5.1.2. Effect of Impinging Distance

Figure 7 shows the simulated behavior of particles at various impinging distances (5 cm and 15 cm), an inlet air velocity of 25 m/s, a feed rate of 160 kg_dry_/h and a temperature of 190 °C. It was seen that the particles moved along the axial direction from the right-hand-side inlet to the impingement zone, and then some particles penetrated the opposite stream. The number of particles penetrated to the opposite stream was increased with the decrease in the impinging distance due to a shorter impinging distance leading to an increase in the acceleration and momentum of particles. The longer inlet pipe (shorter impinging distance) would assist in increasing the acceleration of the particles, leading to a higher particle velocity at the outlet of the right-hand-side inlet pipe. 

Figure 8 shows the simulated particle motion in an impingement zone at various impinging distances (5 cm and 15 cm), an inlet air velocity of 25 m/s, a feed rate of 160 kg_dry_/h and a temperature of 190 °C. It was found that the oscillation of particles took place at impinging distances of 5 cm and 15 cm. However, the oscillated particles at a distance of 5 cm were higher than that of 15 cm due to a higher particle momentum. It is noted that the penetrated distance of particles to the opposite stream increased with a decrease in the impinging distance.

Figure 9a shows a comparison between the simulated moisture content of parboiled paddy particles and experimental data for various impinging distances (5 cm and 15 cm) at an inlet air velocity of 25 m/s, a feed rate of 160 kg_dry_/h and a temperature of 190 °C. They saw that the simulated moisture content decreased faster with a decrease in the impinging distance due to a longer particle residence time (see Table 4) and more turbulence intensity. When the impinging distance decreased, it can be seen that the oscillated particles in the inlet pipe of the stream increased, and hence, the residence time also increased. The tendency of moisture content between simulation and experimental data was the same way. Based on this simulation, the simulated moisture content was higher than that of experimental data at the first cycle of drying. However, the simulated moisture content agreed well with the experimental data after nine cycles of drying. The difference between simulated moisture content and experimental data was approximately 5.2–7.6%.

Figure 9b shows a comparison between simulated particle temperature and experimental data in an impingement zone at various impinging distances (5 and 15 cm) at an inlet air velocity of 25 m/s, a feed rate of 160 kg_dry_/h and a temperature of 190 °C. It was found that the particle temperature increased with an increase in the cycle of drying. The tendency of simulated temperature and experimental data was the same. However, the simulated particle temperature was different from the experimental data, approximately 4.9–5.8%. The discrepancy might be due to the difference in the mean residence time of the simulation and the experimental data.

#### 5.1.3. Feed Rate Effect

Figure 10 shows the simulated behavior of particles at an inlet air velocity of 25 m/s, a temperature of 190 °C, an impinging distance of 5 cm and various particle feed rates of 160 and 320 kg_dry_/h. The particle velocity decreased with an increase in the feed rate because of the increase in particle concentration in the impinging stream inlet, leading to an increase in particle–particle collisions and hence the acceleration of particles at the inlet stream was decreased.

Figure 11 shows the simulated particle motion in an impingement zone at an inlet air velocity of 25 m/s, a temperature of 190 °C, impinging distances of 5 cm and various feed rates. It can be seen that the oscillated particles in the inlet pipe of the stream increased with a decrease in feed rate due to a higher particle velocity, which is a lower feed rate.

Figure 12a shows the comparison between simulation and experimental data for moisture content at different feed rates. Based on the simulation, the moisture content decreased faster as the feed rate decreased due to better moisture transfer to the air at a lower feed rate. Particle–particle collisions were lower at a lower feed rate than at a higher feed rate, allowing the particles to be more exposed to air. The trend of the simulated moisture content and the experimental data was the same. However, the difference between the simulated moisture content and the experimental data was approximately 5.6–7.6%.

Figure 12b shows the comparison between simulation and experimental data for particle temperature at various feed rates. In the simulation, the particle temperature at a feed rate of 160 kg_dry_/h was higher than that of 320 kg_dry_/h, and the tendency of the simulation was the same as that of the experimental data. The discrepancy between the simulation and experimental data was approximately 3.9–5.9%.

#### 5.1.4. Particle Mean Residence Time

Table 4 shows the mean residence time of particles at 190 °C and various parameters of feed rate, air velocity, inlet pipe diameter and impinging distance. Based on the simulation, it can be seen that the mean residence time increased with an increase in air velocity due to a higher of oscillated particles in the impinging stream as previously shown in Figure 5. The mean residence time increased with a decrease in the impinging distance because the oscillated particles increased when the impinging distance decreased. The penetrated distance to the opposite stream also increased with a decrease in the impinging distance, leading to a longer particle mean residence time. In Table 4, it can be seen that the mean residence time increased with a decrease in the particle feed rate. This is because the oscillated particles increased at a lower particle feed rate (see Figure 8). In a comparison between the simulated results and experimental data, it can be seen that the particle mean residence time of simulation was lower than that of the experiment. This might be due to a lower coefficient of restitution (0.6) used in the simulation, which leads to lower particle bouncing, and hence, the mean residence time of particles in the drying chamber decreased.

### 5.2. Design of an ISD

In the previous section, the CFD-DEM, in combination with the falling drying rate model, proved to be a powerful tool for getting detailed information on particle behavior and could be used to predict moisture reduction and particle temperature in ISD. It was noted that the simulated results were in good agreement with the experimental data. For this section, the CFD-DEM, in combination with the falling drying rate model, was applied to design an impinging stream drying chamber to increase particle mean residence time.

#### 5.2.1. Increase of Inlet Pipe Diameter

The particle residence time increased when the momentum of particles increased. To increase particle momentum, the air velocity would be increased, leading to an increase in particle velocity. In the previous section, it is seen that the air velocity of 25 m/s provided a longer particle mean residence time (see Table 4). To prevent erosion of particles and the wall of the drying chamber, the suitable air velocity in an ISD was around 15–30 m/s [42]. Therefore, the air velocity of 25 m/s was used to design the drying chamber. Tamir [1] investigated the ratio between the diameter of the drying chamber (D) and the inlet pipe (d). It was found that an appropriate ratio of D/d was approximately 6; hence, the diameter of the inlet pipe of 0.175 m was used to design the pipe, as shown in Figure 2. It was noted that the diameter of the drying chamber was 1.05 m. 

Figure 13 shows the simulated behavior of particles at an inlet pipe diameter of 0.175 m, an air velocity of 25 m/s, a temperature of 190 °C, a feed rate of 320 kg_dry_/h and various impinging distances. The particle velocity increased with an increase in inlet pipe diameter at the same condition because the momentum of air increased, leading to an increase in particle momentum, and hence, particle velocity increased. As expected, the oscillated particles at a distance of 5 cm were higher than that of 17.5 cm due to a higher particle momentum. The penetrated distance of particles to the opposite stream increased with a decrease in the impinging distance, as shown in Figure 14.

Table 5 shows a large inlet pipe’s particle mean residence time at an air velocity of 25 m/s, a temperature of 190 °C, a feed rate of 320 kg_dry_/h, and impinging distances of 5 and 17.5 cm. It was seen that the mean residence time decreased with an increase in inlet pipe diameter at an impinging distance of 5 cm due to the Venturi effect at the impinging zone, leading to lower pressure and higher air velocity at the stream outlet. Therefore, the particles were easily removed from the impingement zone, causing a lower mean residence time. However, the mean residence time increased when the impinging distance was 17.5 cm because of the disappearance of the Venturi effect at the stream outlet. For moisture content, the moisture reduction increased with an increase in an inlet pipe diameter due to a higher air flow rate, leading to a higher heat transfer between hot air and particles. Although the inlet pipe diameter increased, the mean residence time was slightly changed. However, the moisture reduction of a large inlet pipe ISD (d = 0.175 m) was increased by approximately 3.6 and 4.2 times of an ISD (d = 0.038 m) at impinging distances of 5 and 17.5 cm, respectively. 

#### 5.2.2. Two Stages of Impinging Stream

Tamir [1] investigated the mean residence time of particles in an ISD. It was found that a tangential impinging stream dryer with multistage could increase particle mean residence time. Therefore, this concept was applied to design an ISD with two stages, as shown in Figure 3. The particles would collide with the impinging stream in the first impingement zone (one stage), and then the particles would flow via an inclined wall to collide with the other impinging stream in the second impingement zone (two stages). After that, the particles spread out to a drying chamber.

Figure 15 and Figure 16 show the simulated particle behavior and particle motion in the two stages of ISD. The particles were oscillated in the first stage, then spread out to the inclined wall and flowed into the second stage. In the second stage, the oscillated particles did not occur due to a lower particle velocity. The particles would spread out immediately to a drying chamber after they collided with the opposite stream. The mean residence time of the two stages ISD was increased by approximately 75% when it was compared to a large inlet pipe ISD (d = 0.175 m) and ISD (d = 0.038 m) at impinging distance 5 cm as shown in Table 4 and Table 5. It was noted that the probability distribution of particles and mean residence time of each ISD were shown in Figure 17. For moisture content, the moisture reduction of two stages of ISD increased approximately nine times of an ISD and 2.5 times of a large inlet pipe ISD at the impinging stream distance of 5 cm.

Figure 18a,b show the comparison of simulated moisture content and particle temperature for one-stage and two-stage ISD. The moisture reduction of two-stage ISD was higher than that of one-stage ISD due to a longer mean residence time in two-stage ISD. Therefore, the drying cycle of the two-stage ISD was lower than that of the one-stage ISD at the same final moisture content. It was noted that the one-stage ISD used 17 drying cycles while the two-stage ISD used only 10 drying cycles to reduce the moisture content. As expected, the particle temperature of two-stage ISD was quite higher than that of one-stage ISD because of a longer mean residence time in the two-stage ISD.

## 6. Conclusions

The CFD-DEM, in combination with the falling drying rate model, was used to simulate the particle behavior, moisture content and temperature, as well as the mean residence time of parboiled paddy in a coaxial ISD. The results showed that the CFD-DEM, in combination with the falling drying model, could be utilized to simulate the particle motion behavior, moisture content, temperature, and mean residence time in an ISD. The simulated results of moisture content and temperature of parboiled paddy were in good agreement with the experimental data at various conditions (air velocity, impinging stream distance and feed rate) for 17 cycles of drying. Based on the simulations, the simulated results differed from the experimental data, approximately 4.4–7.6%. However, the simulated mean residence time differed greatly from the experimental data. For ISD design, a large inlet pipe ISD could increase moisture reduction by approximately 3.6–4.2 times of an ISD, whereas there was little change in the mean residence time was a little change. The two-stage ISD could increase the residence time by approximately 75% and reduce the drying cycle from 17 to 10 cycles at the same final moisture content. In addition, the moisture reduction of two stages of ISD increased approximately nine times of an ISD and 2.5 times of a large inlet pipe ISD at the impinging stream distance of 5 cm.

## Figures and Tables

**Figure 1 foods-13-01559-f001:**
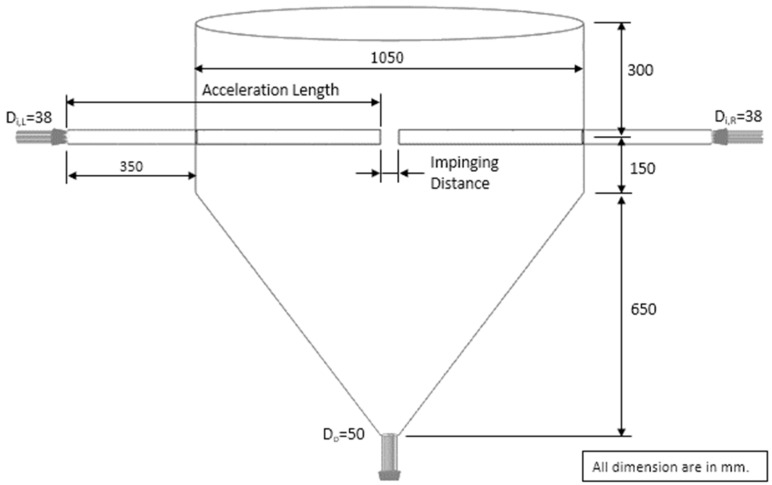
Schematic diagram of the simulated ISD with small inlet pipes.

**Figure 2 foods-13-01559-f002:**
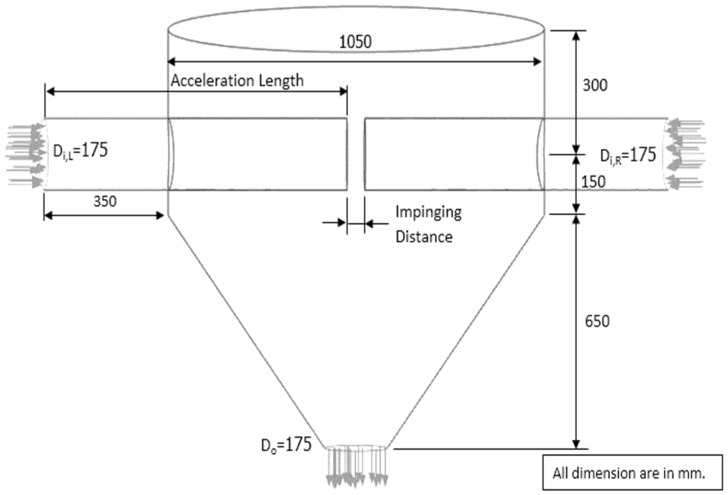
Schematic diagram of the simulated ISD with large inlet pipes.

**Figure 3 foods-13-01559-f003:**
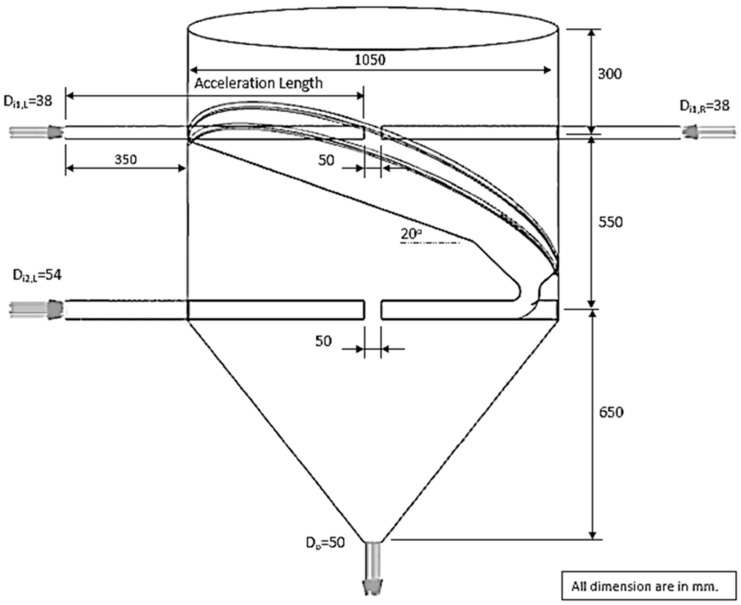
Schematic diagram of the simulation ISD with two stages.

**Figure 4 foods-13-01559-f004:**
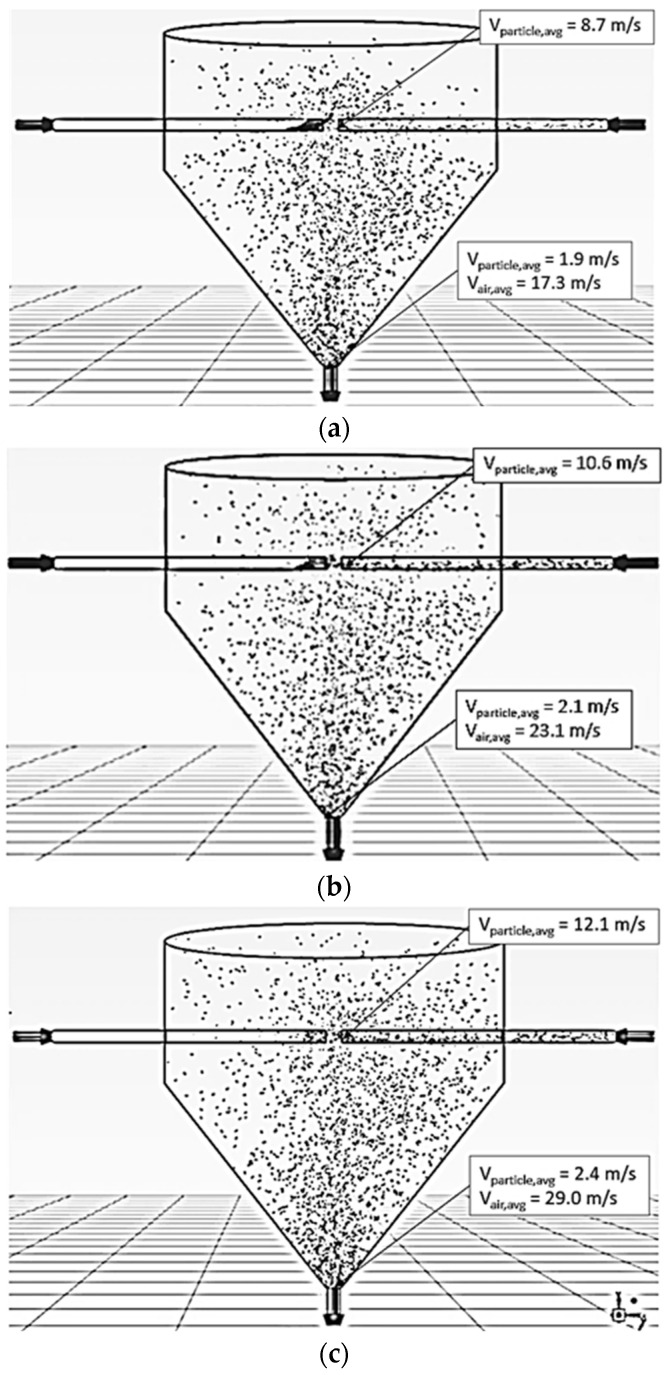
Simulated behavior of particle at a feed rate of 320 kg_dry paddy_/h, temperature of 190 °C, impinging distance of 5 cm and various inlet air velocities: (**a**) 15 m/s (**b**) 20 m/s (**c**) 25 m/s.

**Figure 5 foods-13-01559-f005:**
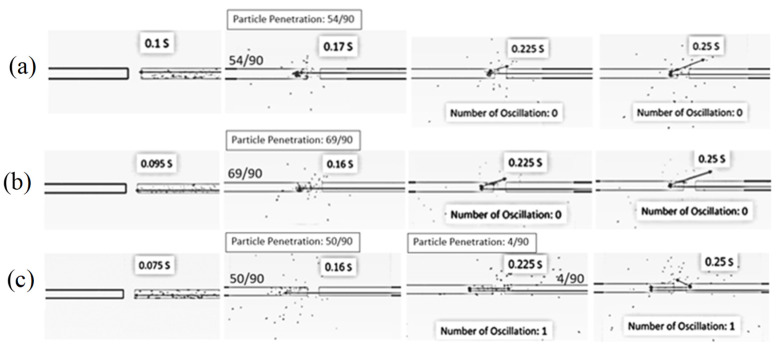
Simulated particle motion in an impingement zone at a feed rate of 320 kg_dry paddy_/h, temperature of 190 °C, impinging distance of 5 cm and various inlet air velocities: (**a**) 15 m/s (**b**) 20 m/s (**c**) 25 m/s.

**Figure 6 foods-13-01559-f006:**
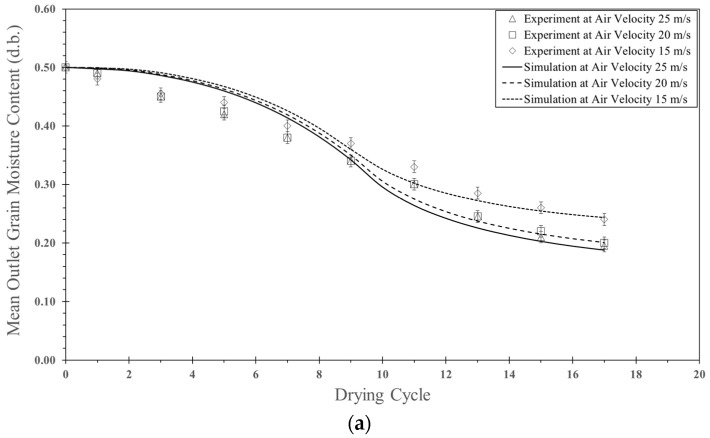
Comparison between simulation and experimental data at various air velocities (15, 20, and 25 m/s), temperature of 190 °C feed rate of 320 kg_dry paddy_/h and impinging distance of 5 cm: (**a**) moisture content and (**b**) particle temperature.

**Figure 7 foods-13-01559-f007:**
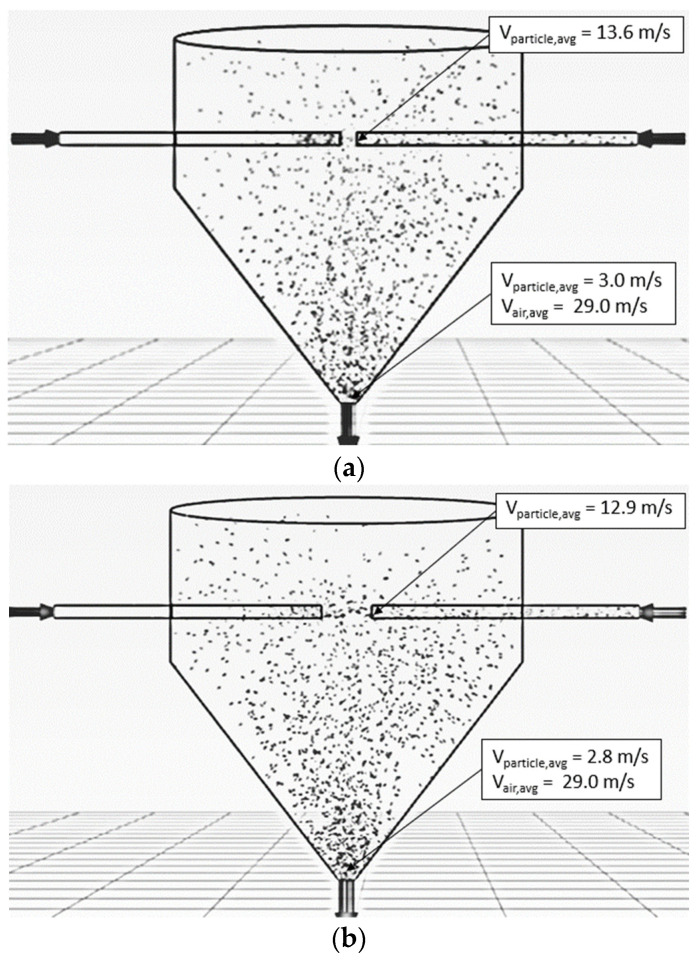
Simulated behavior of particle at inlet air velocity of 25 m/s, feed rate of 160 kg_dry paddy_/h, temperature of 190 °C and various impinging distances: (**a**) 5 cm (**b**) 15 cm.

**Figure 8 foods-13-01559-f008:**
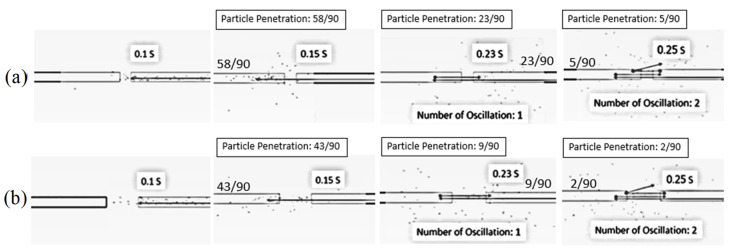
Simulated particle motion in an impingement zone at inlet air velocity of 25 m/s, feed rate of 160 kg_dry paddy_/h, temperature of 190 °C and various impinging distances: (**a**) 5 cm (**b**) 15 cm.

**Figure 9 foods-13-01559-f009:**
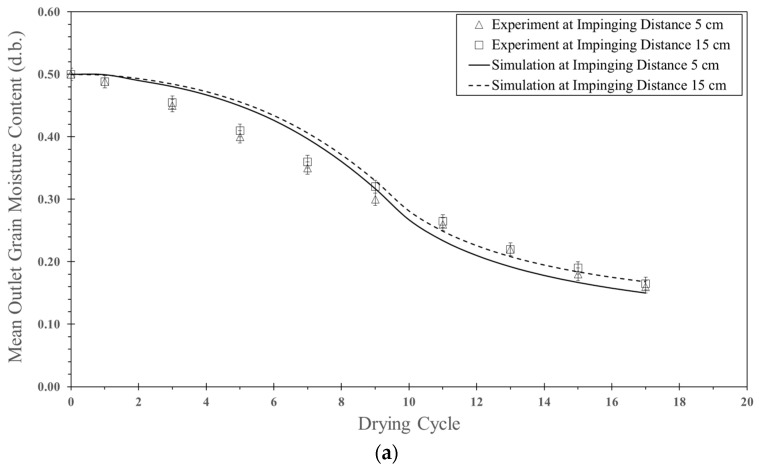
Comparison between simulation and experimental data at various impinging distances (5 and 15 cm); inlet air velocity of 25 m/s; temperature of 190 °C; feed rate of 160 kg_dry paddy_/h: (**a**) moisture content and (**b**) particle temperature.

**Figure 10 foods-13-01559-f010:**
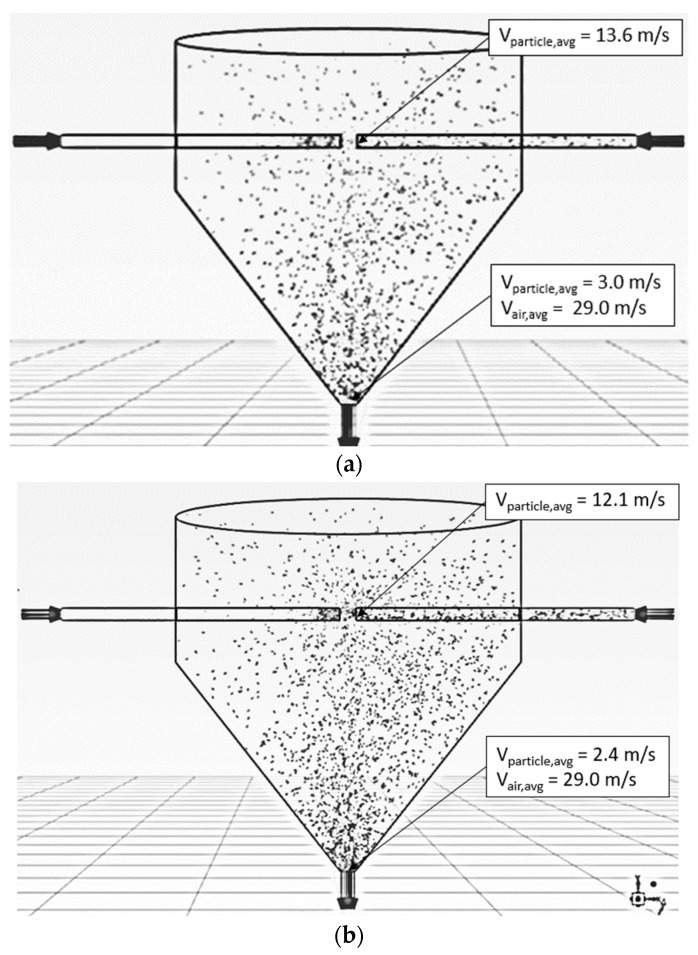
Simulated particle behavior at an inlet air velocity of 25 m/s, temperature of 190 °C, impinging distances of 5 cm and various feed rates: (**a**) 160 kg_dry paddy_/h (**b**) 320 kg_dry_
_paddy_/h.

**Figure 11 foods-13-01559-f011:**
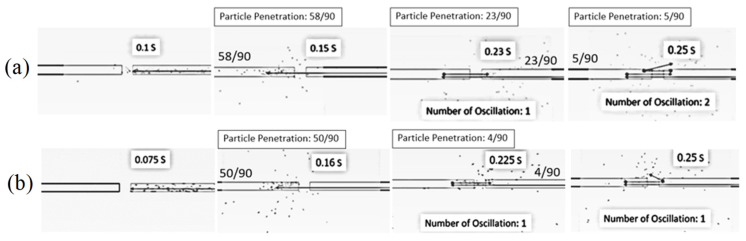
Simulated particle motion in an impingement zone at an inlet air velocity of 25 m/s, temperature of 190 °C and impinging distances of 5 cm and various feed rates: (**a**) 160 kg_dry paddy_/h (**b**) 320 kg_dry paddy_/h.

**Figure 12 foods-13-01559-f012:**
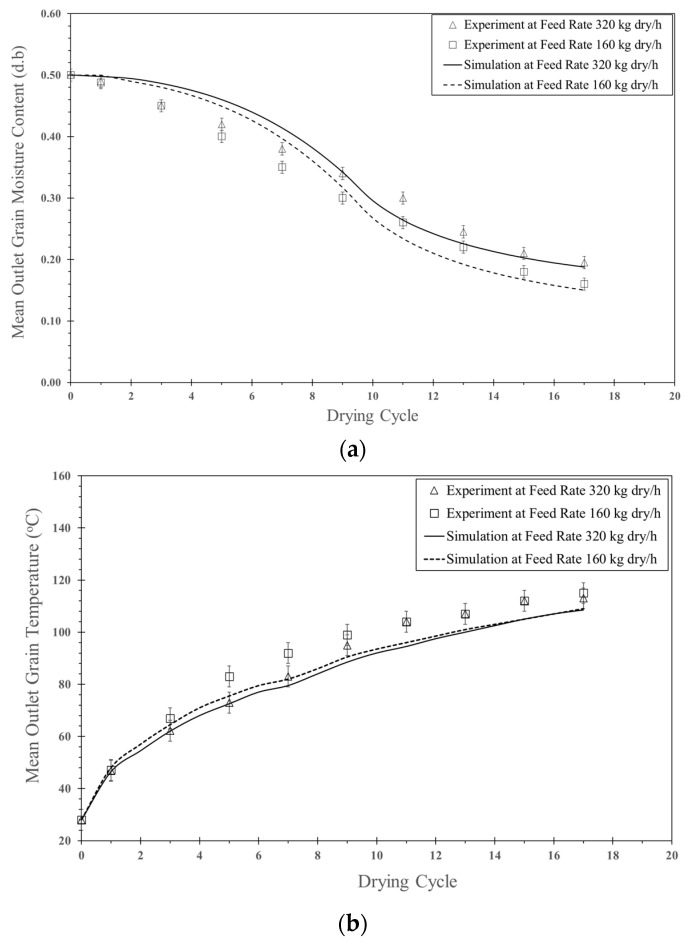
Comparison between simulation and experimental data at various feed rates (160 and 320 kg_dry paddy_/h), inlet air velocity of 25 m/s, temperature of 190 °C and impinging distance of 5 cm: (**a**) moisture content and (**b**) particle temperature.

**Figure 13 foods-13-01559-f013:**
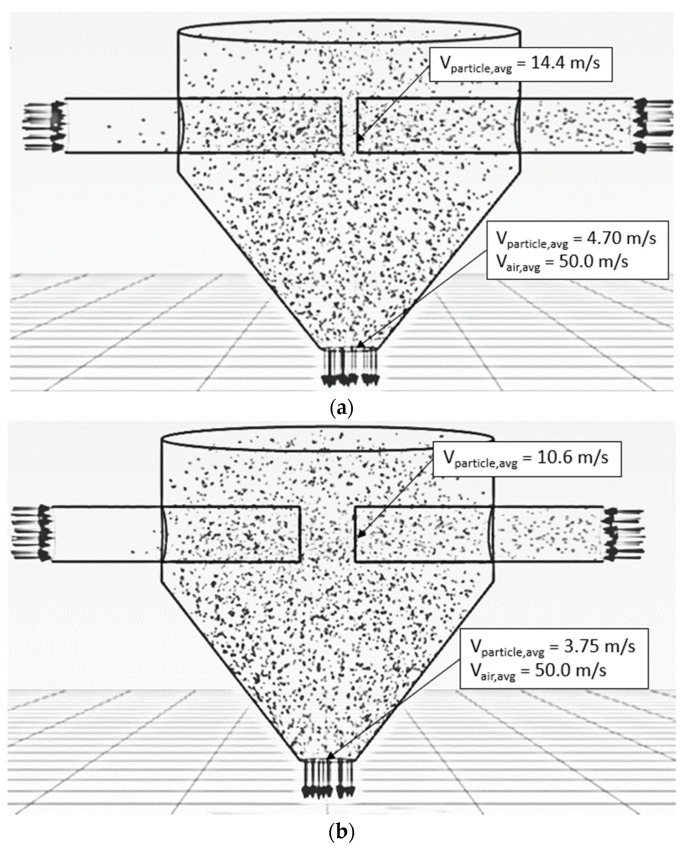
Simulated behavior of particle at an inlet pipe diameter of 0.175 m, air velocity of 25 m/s, temperature of 190 °C, feed rate of 320 kg_dry paddy_/h and various impinging distances: (**a**) 5 cm; (**b**) 17.5 cm.

**Figure 14 foods-13-01559-f014:**
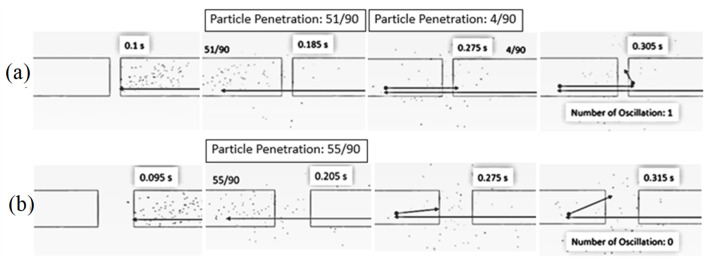
Simulated particle motion at an inlet pipe diameter of 0.175 m, air velocity of 25 m/s, temperature of 190 °C, feed rate of 320 kg_dry paddy_/h and various impinging distances: (**a**) 5 cm; (**b**) 17.5 cm.

**Figure 15 foods-13-01559-f015:**
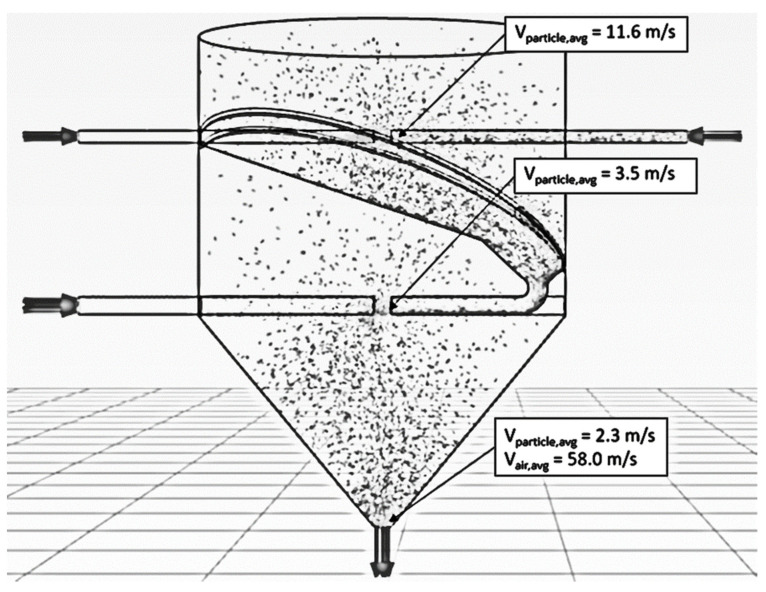
Simulated particle behavior in an ISD with two stages at an air velocity of 25 m/s, with a temperature of 190 °C and a feed rate of 320 kg_dry paddy_/h.

**Figure 16 foods-13-01559-f016:**
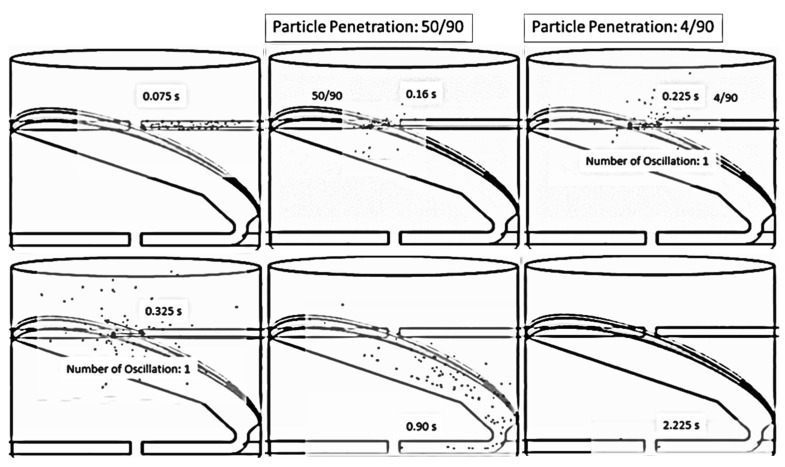
Simulated particle motion in an ISD with two stages at an air velocity of 25 m/s, with a temperature of 190 °C and a feed rate of 320 kg_dry paddy_/h.

**Figure 17 foods-13-01559-f017:**
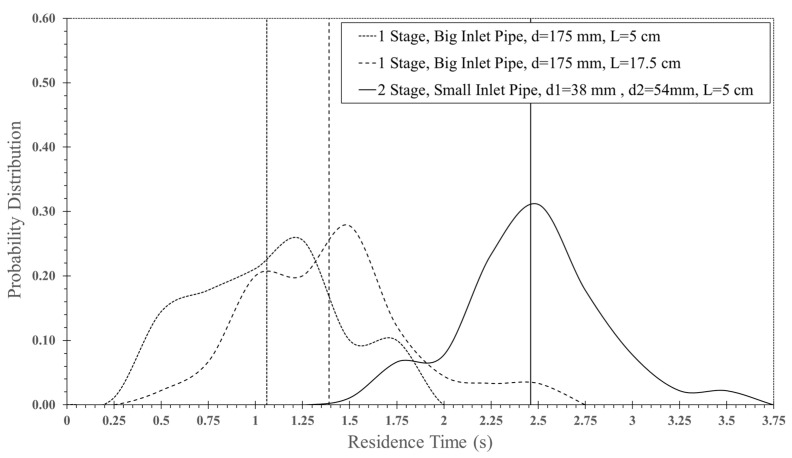
Probability distribution of particles and mean residence time of each ISD.

**Figure 18 foods-13-01559-f018:**
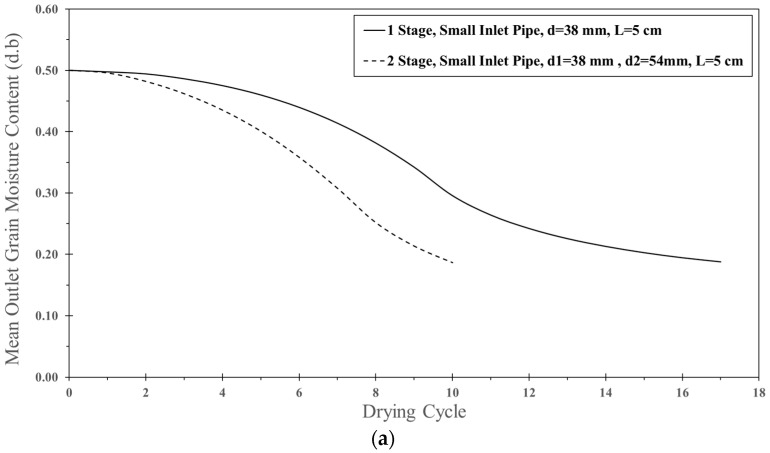
Comparison between simulation of one-stage and two-stage ISD at feed rates 320 kg_dry paddy_/h), inlet air velocity of 25 m/s, temperature of 190 °C, and impinging distance of 5 cm: (**a**) moisture content and (**b**) particle temperature.

**Table 1 foods-13-01559-t001:** Stable time of particles at different operating parameters.

Feed Rate (kg dry/h)	Air Temperature (°C)	Air Velocity (m/s)	Inlet Pipe Diameter (mm)	Impinging Distance (cm)	Stable Time (s)
320	190	15	38	5	1.68
	190	20	38	5	1.70
	190	25	38	5	1.83
160	190	25	38	5	2.17
	190	25	38	15	1.90
320	190	25	175	5	1.90
	190	25	175	17.5	2.17
	190	25	First Stage: 38, Second Stage: 54	5	3.10

**Table 2 foods-13-01559-t002:** Properties of particle phase in simulation.

Properties	Correlation Used in Simulation
Diameter (m) [5]	0.0039
Density (kg/m^3^) [35]	(1835 M_db_) + 487.03
Heat capacity (kJ/kg⋅K) [36]	1.1188 + (5.8362 × 10^−3^ T_p_) + (3.4695 × 10^−2^ M_db_) − (1.3432 × 10^−4^ T_p_ M_db_) − (2.4808 × 10^−4^ M_db_^2^)
Thermal conductivity (W/m⋅K) [37]	(0.0637 + 0.0958 (M_db_/(M_db_ + 1)))/ (0.656 − 0.475 (M_db_/(M_db_ + 1)))
Heat of vaporization (kJ/kg) [36,38]	(2502 − (2.386 T_p,c_)) × (1 + (2.496 (e^−21.733·Mdb^)))
Spring constant (N/m) [17,26,27,28]	1000
Restitution coefficient [29]	
Particle–particle	0.6
Particle–wall	0.6
Initial moisture content (d.b.) [5]	0.5
Arrhenius factor (m^2^/s) [39]	2.55 × 10^−7^
Activation energy (J/mol) [39]	20,580

**Table 3 foods-13-01559-t003:** Properties of the gas phase in simulation [34].

Properties	Correlation Used in Simulation
Thermal conductivity (W/m⋅K)	(1.3 × 10^−3^) + ((9.11 × 10^−5^)T_a_) − ((2.52 × 10^−8^)T_a_^2^)
Heat capacity (kJ/kg⋅K)	990 − ((1.77 × 10^−2^)T_a_) + ((1.91 × 10^−4^)T_a_^2^)
Viscosity (Pa⋅s)	(3.53 × 10^−6^) + ((5.54 × 10^−8^)T_a_) + ((1.70 × 10^−11^)T_a_^2^)

**Table 4 foods-13-01559-t004:** Mean residence time of particles at different operating parameters.

Feed Rate (kg dry/h)	Air Temperature (°C)	Air Velocity (m/s)	Inlet Pipe Diameter (mm)	Impinging Distance (cm)	Mean Residence Time (s)		
					Simulation	Experiment [5,41]	Percent Relative Error
320	190	15	38	5	1.26	1.4 ± 0.18	10.0%
	190	20	38	5	1.38	1.8 ± 0.20	23.3%
	190	25	38	5	1.40	2.2 ± 0.20	36.4%
160	190	25	38	5	1.53	2.6 ± 0.12	41.2%
	190	25	38	15	1.32	2.4 ± 0.15	45.0%
320	190	25	175	5	1.06	N/A	N/A
	190	25	175	17.5	1.39	N/A	N/A
	190	25	First Stage: 38, Second Stage: 54	5	2.46	N/A	N/A

**Table 5 foods-13-01559-t005:** Mean residence time of particles at air velocity of 25 m/s, temperature of 190 °C, feed rate of 320 kg_dry paady_/h and impinging distances of 5 and 17.5 cm for various inlet pipe diameters and two stages of impinging stream.

Feed Rate (kg dry/h)	Air Temperature (°C)	Impinging Distance (cm)	Air Velocity (m/s)	No. of Stage	D (m)	D/d	d (m)	1st Drying Cycle (M_in_ = 0.5 d.b.)		
								Mean Residence Time (s)	Mean Outlet Moisture (d.b.)	∆M (d.b)
320	190	5	25	1	1.05	27.6	0.038	1.40	0.4995	0.0005
320	190	5	25	1	1.05	6	0.175	1.06	0.4982	0.0018
320	190	17.5	25	1	1.05	6	0.175	1.39	0.4979	0.0021
320	190	5	25	2	1.05	27.6 ^1^*	0.038 ^1^*	2.46	0.4955	0.0045
						19.5 ^2^*	0.054 ^2^*			

1* is the parameter of the first impinging stream in the drying chamber (first stage). 2* is the parameter of the second impinging stream in the drying chamber (second stage).

## Data Availability

The original contributions presented in the study are included in the article, further inquiries can be directed to the corresponding author.

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
