# Peer review of "Mathematical Modeling and Design of Parboiled Paddy-Impinging Stream Dryer Using the CFD-DEM Model"

_foods, 2024, doi:10.3390/foods13101559_

Round 1

Reviewer 1 Report

Comments and Suggestions for Authors

The paper: Mathematical Modeling and Design of Parboiled Paddy Impinging Stream Dryer using CFD-DEM model, by authors: Thanit Swasdisevi ,Wut Thianngoen and  Somkiat Prachayawarakorn presents numerical investigation of the drying process with numerical method CFD/DEM. The authors performed a detail numerical experiments and  the model was very demanding. But in my opinion, the presentation of the results could be improved.

Comments:

1.       Please, add information about the application and advantages of the impinging stream dryer in the Introduction section.

2.       Also, in the introduction section add the novelty of your research and the significance of your method.

3.       For interphase model you used several correlations. Please, explain why you used these correlations, and not some others. (application of the model)

4.       You used k-e turbulence model. Why? Please, add the advantages of this model.

5.       Check the equation 18 or explain the coefficient.

6.       Please, improve the quality of figures. I am sure you can do this in Ansys software, especial fig. 5, 8, 11, 14. All figures can be improved. If this is not possible, please, explain.

7.       The quality of the letters used in figures and equations can be improved.

Comments on the Quality of English Language

/

Author Response

  1. Please, add information about the application and advantages of the impinging stream dryer in the Introduction section.

This information has been already specified. See line 33-40 (foods-2972857) for ISD fundamental. See line 41-48 (foods-2972857) for application comparing between ISD dryer and other dryer.

2. Also, in the introduction section add the novelty of your research and the significance of your method.

This information has been already added and specified. See line 91-95 (foods-2972857) for applying the falling drying rate concept in the model. This is quite new and needs a lot of effort to apply by UDF in Ansys(Fluent).

3. For interphase model you used several correlations. Please, explain why you used these correlations, and not some others. (application of the model)

In the momentum equation, we used Wen and Yu correlations because these correlations were suitable for our work. The details of the correlations are given in the reference 12 and 20.

4) You used k-e turbulence model. Why? Please, add the advantages of this model.

A realizable k-e turbulence model is suitable for high turbulence systems like ISD dryer and the information was added (see line 126).

5) Check the equation 18 or explain the coefficient.

This is the energy source in gas phase control volume with transferring energy(heat) to Parboiled Paddy (see lines 165-166).

  1. Please, improve the quality of figures. I am sure you can do this in Ansys software, especial fig. 5, 8, 11, 14. All figures can be improved. If this is not possible, please, explain.

This is high quality pictures which we got for current version of Ansys(Fluent). This particle movement came from we snapshot 2-times (1st time for Frame and 2nd time for particle movement). Then, we will combine frame and particle movement.

Reviewer 2 Report

Comments and Suggestions for Authors

The article “Mathematical Modeling and Design of Parboiled Paddy Impinging Stream Dryer using CFD-DEM model” has been reviewed and the following comments have been made to improve the manuscript

The topic is novel and relevant from the point of view of fundamental analysis.

I consider that these “emerging” technologies for food drying are of commercial importance. The introduction is adequate and clearly shows the state of the art of the problem.

The last paragraph of the introduction should be clearer in the sense of establishing the general objective of the research and its scope.

The equations based on them are adequate. Figure 1 will be able to stand out better if it is accompanied by a real photograph of the system.

The methodology should be a little more descriptive for those readers who are introducing themselves to the topic. It is important to describe in detail what the drying process is like, how much sample was selected, what each drying cycle consists of, how long each cycle lasts, what is the range of speeds selected. An abstract graphic can help understand the methodology they followed.

Line 222. Specify Kg what you are talking about

Figure 5 needs to improve sharpness.

Figure 6 (b) modify °C units

There is no table showing the statistical contrast results of the experimental part with the simulated part, % error in addition to other indicators.

How would the drying rate, effective diffusivity and its relationship with time, residence, temperature and air speed be explained. This type of analysis is important to be able to weight the effect of each of the factors evaluated in your model.

The conclusions must relate to each of the experiments evaluated and make a global comparison. It is also important to generate expectations for future work or the implications of this work in possible future applications.

Author Response

Comments and Suggestions for Authors
The article “Mathematical Modeling and Design of Parboiled Paddy Impinging Stream Dryer using CFD-DEM model” has been reviewed and the following comments have been made to improve the manuscript
The topic is novel and relevant from the point of view of fundamental analysis.
I consider that these “emerging” technologies for food drying are of commercial importance. The introduction is adequate and clearly shows the state of the art of the problem.
1) The last paragraph of the introduction should be clearer in the sense of establishing the general objective of the research and its scope.
We have already made a change in the last paragraph.
2) The equations based on them are adequate. Figure 1 will be able to stand out better if it is accompanied by a real photograph of the system.
Thank you for your good suggestion. Sorry, we do not have a real photograph of the ISD since we have removed a pilot scale ISD and installed for a lab scale ISD for doing the research.
3) The methodology should be a little more descriptive for those readers who are introducing themselves to the topic. It is important to describe in detail what the drying process is like, how much sample was selected, what each drying cycle consists of, how long each cycle lasts, what is the range of speeds selected. An abstract graphic can help understand the methodology they followed.
The brief methodology was added in the manuscript (see line 102-112)
4) Line 222. Specify Kg what you are talking about
“We did not find Kg at line 222. However, if you mean “kc, it presented the Air t hermal conductivity.
5) Figure 5 needs to improve sharpness.
This is high quality picture which got from the current version of Ansys(Fluent). This particle movement coming from snapshot 2-times (1st time for Frame and 2nd time for particle movement). Then, we combined frame and particle movement.
6) Figure 6 (b) modify °C units
We have already done.
7) There is no table showing the statistical contrast results of the experimental part with the simulated part, % error in addition to other indicators.
We have already added the percent error of mean residence time in Table 4. (foods-2972857).

Round 2

Reviewer 1 Report

Comments and Suggestions for Authors

The paper, in the present form can be published.

Comments on the Quality of English Language

/